

# Blowing snow contributions to the Arctic snow-on-sea ice budget using ICESat-2 observations

Joseph Robinson[1], Lyatt Jaeglé[1], Stephen P. Palm[2,3], Glen E. Liston[4]

[1] Department of Atmospheric and Climate Science, University of Washington, Seattle, WA, USA
[2] Science Systems and Applications, Lanham, MD, USA
[3] NASA Goddard Space Flight Center, Greenbelt, MD, USA
[4] Cooperative Institute for Research in the Atmosphere (CIRA), Colorado State University, Fort Collins, CO, USA

*Correspondence to:* Joseph Robinson ([jrobin15@uw.edu](mailto:jrobin15@uw.edu))





**Abstract**

Blowing snow modulates the evolution of snow over Arctic sea ice through redistribution and sublimation. Here, we present the first multi-year pan-Arctic observational estimates of blowing snow occurrence, properties, and associated fluxes based on NASA Ice, Cloud and land Elevation Satellite 2 (ICESat-2) satellite observations for five cold seasons (November through April 2018-2023). On average, ICESat-2 detects blowing snow 19% of the time over sea ice, with localized frequencies reaching up to 35% in the Central Arctic, where blowing snow heights (optical depths) reach 150 m (0.20). We find that blowing snow occurrence shows strong interannual variability related to large-scale climate variability, particularly the Arctic Oscillation (AO). During positive AO phases, blowing snow occurrence increases substantially, with up to a two-fold increase in the Central Arctic. Blowing snow occurrence, height, and optical depth all exhibit a strong dependence on wind speed, increasing by more than five-fold between 4 and 15 m s$^{-1}$. ICESat-2 blowing snow sublimation estimates average 1.63 cm snow-water-equivalent (SWE) per cold season, thus removing 14% of pan-Arctic snowfall. In the Central Arctic, the offset is 18-24%. These values are consistent with simulations from the high-resolution SnowModel-LG (1.66 cm SWE) and a simpler, threshold-based model (2.07 cm SWE). Interannual variability in snowfall and sublimation can be 1-2 cm SWE, though not always in phase, resulting in snowfall removals that range from 9% to 20%. Critically, these findings provide satellite-based constraints on blowing snow processes over sea ice and underscore the importance of blowing snow sublimation in the Arctic snow budget.

**1 Introduction**

Snow cover on sea ice is a fundamental component of the Arctic climate system, influencing surface albedo, insulating the ocean from the atmosphere, and modulating the exchange of heat and moisture across the ocean-ice-atmosphere interface (Merkouriadi, Cheng, et al., 2017; Merkouriadi, Gallet, et al., 2017; Sturm et al., 2002; Webster et al., 2018). Its presence impacts not only the local energy balance but also broader climate feedbacks that affect both high- and mid-latitudes. As the Arctic undergoes rapid environmental change, including thinning sea ice (Kwok & Untersteiner, 2011; Stroeve & Notz, 2018), shifting precipitation patters (Bintanja, 2018; Bintanja & Andry, 2017; McCrystall et al., 2021), and increasing temperatures (Rantanen et al., 2022), the need to accurately characterize the spatial and temporal variability of snow on sea ice has become increasingly urgent. Understanding how the snowpack and its properties evolve across a range of spatial and temporal scales and in response to dynamic atmospheric processes is essential for improving predictions of sea ice behavior, refining climate model simulations, and assessing implications for Arctic ecosystems, human activities, and global climate.

While the snowpack on sea ice generally follows a seasonal cycle of winter accumulation and summer melt, shorter-term processes can alter characteristics and accumulation rates. One such process is blowing snow, which occurs when strong winds lift snow away from the surface. Numerous studies spanning several decades have underscored the role of blowing snow in modulating sea and land ice mass balance (Déry & Yau, 2002; Gallée et al., 2001; Palm et al., 2017), altering radiative properties in polar regions (Lesins et al., 2009; Y. Yang et al., 2014), impacting chemical processes in the polar troposphere (Frey et al., 2020; Gong et al., 2023;




Huang et al., 2020; Huang & Jaeglé, 2017; Krnavek et al., 2012; X. Yang et al., 2008), and
complicating the interpretation of physical and chemical ice core records (King et al., 2004;
Rhodes et al., 2017). Yet, capturing the full spatial and temporal variability of blowing snow
remains challenging due to the limited availability of sustained, regionally comprehensive
observations (Déry & Yau, 2001; Mann et al., 2000; Nishimura & Nemoto, 2005).
When lifted into the air, blowing snow particles are exposed to conditions that can promote their
sublimation, making blowing snow sublimation a significant pathway for both snow removal and
a source of atmospheric moisture. While sublimation can occur directly at the snow surface, it is
far more efficient when particles are suspended aloft, where their full surface area interacts with
the ambient air (Liston & Sturm, 2004; Schmidt, 1982). Model-based assessments suggest a
substantial role for this process in the Arctic hydrological cycle: J. Yang et al. (2010) estimated
that over 27% of winter snowfall poleward of 70°N may be lost to blowing snow sublimation.
However, other modeling studies (e.g., Chung et al., 2011; Déry & Yau, 2002) have reported
much lower estimates (6-7%), underscoring the considerable uncertainty that still surrounds
blowing snow related processes. Narrowing these uncertainties and understanding the
implications of sublimation-driven snow loss over sea ice remains a pressing scientific challenge.
The time evolution of snow-water-equivalent (SWE) depth can be described by the mass balance
equation:

$$\frac{dSWE}{dt} = \frac{1}{\rho_w}[P - (M + Q_{ss} + Q_{bs}) + Q_t] \tag{1}$$

where $\rho_w$ is the density of water, and the terms represent inputs from precipitation ($P$; kg m$^{-2}$ s$^{-1}$)
and losses via melt ($M$, kg m$^{-2}$ s$^{-1}$) and sublimation (kg m$^{-2}$ s$^{-1}$), either from a static, non-blowing
snow surface ($Q_{ss}$) or via blowing snow ($Q_{bs}$). Erosion and deposition by blowing snow transport
($Q_t$, kg m$^{-2}$ s$^{-1}$) can also play a role in shaping the local snowpack. While Eq. 1 represents key
drivers of snowpack evolution, other processes, such as ice dynamics (e.g., creation and
destruction of parcels through ice motion, divergence, and convergence), may also play
important roles.
Efforts to quantify the influence of blowing snow on SWE often rely on empirical
parameterizations of snow transport and sublimation derived from sparse observations. These
approaches typically use meteorological inputs such as windspeed, air temperature, and snow age
to estimate thresholds for blowing snow initiation and subsequent sublimation (e.g., Gallée et al.,
2001, 2013; Lenaerts et al., 2010, 2012). In the Northern Hemisphere, model development has
primarily focused on continental snowpacks (Déry & Yau, 2001, 2002; Pomeroy et al., 1997; J.
Yang & Yau, 2007), where snow redistribution is critical to understand human relevant
hydrology and impacts to infrastructure. Although several studies have extended these
approaches to sea ice environments (Chung et al., 2011; Déry & Tremblay, 2004; Lecomte et al.,
2015; Liston et al., 2018, 2020; J. Yang et al., 2010), there remains a lack of direct, observation-
based constraints on pan-Arctic blowing snow processes over sea ice.
Spaceborne lidars offer a powerful means to address observational gaps and assess the
occurrence and impacts of blowing snow across large spatial and temporal domains. Palm et al.
(2011, 2017, 2018) developed a detection algorithm for the Cloud-Aerosol Lidar with



Orthogonal Polarization (CALIOP) aboard NASA's CALIPSO satellite (Winker et al., 2009),
demonstrating that lidar backscatter measurements can be used to quantify key blowing snow
characteristics over Antarctica, including frequency of occurrence, height, optical depth, and
associated transport and sublimation fluxes. Building on this approach, a similar algorithm was
later adapted for the NASA Ice, Cloud, and land Elevation Satellite-2 (ICESat-2; Markus et al.,
2017) by Palm et al. (2021) and Herzfeld et al. (2021). Both algorithms were tailored to detect
blowing snow over the Antarctic continent. More recently, Robinson et al. (2025) optimized the
ICESat-2 blowing snow detection algorithm for application over Arctic sea ice, where more
frequent low-level cloud cover (Shupe et al., 2011; Zhang et al., 2019) increases the likelihood of
both false positives (i.e., clouds misidentified as blowing snow) and false negatives (i.e., blowing
snow misclassified as clear air) in lidar retrievals. Robinson et al. (2025) demonstrated that
retrieval errors caused by cloud interference can be effectively corrected, enabling the
development of a space-based blowing snow detection product specifically adapted for Arctic
sea ice.
In this study, our goal is to examine blowing snow occurrence and properties inferred from
ICESat-2 over Arctic sea ice across five cold seasons (defined as November through April)
between 2018 to 2023. We use ICESat-2 observations to infer blowing snow sublimation and its
role in the snow-on-sea ice budget. We compare the ICESat-2 observations to blowing snow
simulations from two models of varying complexity: a parameterization based on the PIEKTUK
blowing snow model (DY2001; Déry & Yau, 1999, 2001; J. Yang & Yau, 2007) and the state-of-
the-art Lagrangian snow-evolution model SnowModel-LG (Liston et al., 2020).
In Section 2 we provide details on the ICESat-2 blowing snow retrievals and inferred blowing
snow properties, SnowModel-LG predictions, and the DY2001 blowing snow sublimation
formulation. In Section 3 we present the ICESat-2 multi-year blowing snow occurrence
frequency and properties, examining key drivers of their spatiotemporal distribution. The role of
blowing snow in the snow-on-sea-ice budget is examined in Section 4 and conclusions are
presented in Section 5.
**2 Datasets and Methods**
**2.1 Satellite blowing snow retrievals from ICESat-2**
ICESat-2 was launched in 2018 in a precessing orbit with an altitude of ~ 500 km and inclination
of 92°, which allows for measurements up to 88° N latitude with a 91-day orbital repeat cycle
(Markus et al., 2017). ICESat-2 carries the Advanced Topographic Laser Altimeter System
(ATLAS), which is a single wavelength (532 nm), high repetition rate (10 kHz) lidar system with
photon counting detectors (Markus et al., 2017; Neumann et al., 2019). Each ATLAS laser pulse
is split into 3 simultaneous beam pairs (one strong and one weak beam per pair) by a diffractive
optical element. The 3 beam pairs are separated by about 3 km across track. Atmospheric
backscatter is obtained by ATLAS using only the three strong beams, spanning from the surface
to an altitude of 14 km, with an along-track resolution of approximately 280 m and a vertical
resolution of 30 m. Each 280 m ICESat-2 atmospheric profile represents the aggregate of 400
individual ATLAS laser shots (Palm et al., 2021). In this study we use ICESat-2 strong beam 1
observations from version 6 of the ATLAS/ICESat-2 Level 3A (ATL09) calibrated backscatter
profile product (Palm et al., 2023).



The algorithm used to detect blowing snow in ATLAS backscatter profiles is adapted from the
CALIOP approach (Palm et al., 2011) and further detailed in Palm et al. (2021; 2022). When a
surface return is identified and the 10 m wind speed from NASA's GEOS-5 FP-IT analysis
exceeds 4 m s$^{-1}$, the algorithm compares the near-surface atmospheric backscatter to the expected
molecular (Rayleigh) signal. If the observed signal exceeds a fixed multiple of the molecular
scattering, the algorithm steps upward through each vertical bin until the backscatter drops below
an adaptive threshold (typically $\sim 2\times10^{-5}$ m$^{-1}$ sr$^{-1}$). To be flagged as blowing snow, the detected
feature must touch the ground and be shallower than 500 m. Retrievals deeper than 500m are
classified as diamond dust, which can stretch for a km or more vertically and frequently reaches
the ground (Intrieri & Shupe, 2004). Further, we use the version of the blowing snow algorithm
described in Robinson et al. (2025) which includes modifications to help alleviate several
challenges unique to the Arctic. These modifications serve to 1) minimize the misidentification
of low clouds as blowing snow and 2) correct for the attenuation due to transmissive clouds.
Once blowing snow is retrieved, its properties (geometric and optical depths) are logged. Optical
depth (OD) is estimated as the sum of the backscatter within the blowing snow retrieval
multiplied by the product of the bin depth (30 m) and the extinction to backscatter (lidar) ratio. A
lidar ratio of 25 sr is used, which is a typical value for ice crystals in cirrus clouds (Chen et al.,
2002; Josset et al., 2012). To infer blowing snow particle number density, transport flux, and
sublimation flux from the observed ICESat-2 backscatter we follow the same approach as
described in Palm et al. (2017) and Robinson et al. (2025), which relies on meteorological fields
(10 m wind speed, 2 m temperature, and 2 m relative humidity over ice) from the NASA GEOS-
5 FP-IT analysis (run at 0.5$^{\circ}$ latitude × 0.625$^{\circ}$ longitude; Lucchesi et al., 2015) as well as
assumptions about blowing snow particle size. As in Robinson et al. (2025) we use the
formulation $r(z)=5.05\times10^{-5} z^{-0.085}$ to estimate the particle radius ($r$, meters) as a function of
altitude ($z$, meters). This fit was constrained by observations of blowing snow particle sizes
during the 2019-2020 Multidisciplinary drifting Observatory for the Study of Arctic Climate
(MOSAiC) campaign.
To improve signal-to-noise in sunlit conditions, we apply along-track averaging to the ICESat-2
observations when the solar elevation angle exceeds -7°, a threshold beyond which background
solar photons begin to significantly degrade sensitivity. Under these conditions, which affect late
February through April (Fig. S1), increased solar background can reduce the detectability of low-
backscatter features such as blowing snow. To mitigate this, we average the native 25 Hz (280
m) profiles to 1 Hz ($\sim$7 km) resolution, effectively reducing solar background noise and
enhancing the reliability of blowing snow retrievals. While this approach lowers spatial
resolution, it reduces false positive detections and provides a more robust estimate of blowing
snow properties under marginal lighting conditions without introducing significant biases in
seasonal statistics.
**2.2 Blowing snow model simulations from SnowModel-LG**
SnowModel-LG is a physics-based snow-on-sea ice model forced by atmospheric inputs of air
temperature, RH, winds, and precipitation by the NASA Modern-Era Retrospective analysis for
Research and Applications, version 2 (MERRA-2; Gelaro et al., 2017) as well as sea ice inputs





of concentration and parcel motion (Tschudi et al., 2019, 2020). At each 3-hour timestep,
SnowModel-LG performs mass-budget calculations (e.g., Eq. 1) where SWE depth evolution is
accounted for by snow gains, losses, and sea ice dynamics (Liston et al., 2020).
The MicroMet module (Liston & Elder, 2006) is used to time average (1-hourly to 3-hourly) and
distribute the MERRA-2 fields ($0.5^o$ latitude × $0.625^o$ longitude) to the sea ice parcels. As part of
this procedure, the MERRA-2 water equivalent precipitation is bias corrected (as described in
section 2.5 and Table 1 of (Liston et al., 2020) and partitioned into snowfall and rainfall based on
environmental conditions (Dai, 2008).
Blowing snow in SnowModel-LG is accounted for by SnowTran-3D (Liston et al., 2007, 2018;
Liston & Sturm, 1998). The snow threshold friction velocity, $u_{*t}$, is calculated as a function of
snow density, $\rho_s$, which is related to snow strength and hardness. Snow density evolution
includes the history of temperature, precipitation, and wind-transport. When the friction velocity
exceeds the threshold value, snow begins to be lifted off the surface, first into the saltation layer
(several cm thick) and then into the turbulent suspension layer (several m thick). The vertical
mass concentration in the blowing snow profile is estimated following Liston & Sturm (1998)
and is combined with the environmental conditions to calculate transport and sublimation fluxes.
Mass transport is related to the windspeed and vertical mass concentration. SnowModel-LG's
blowing snow sublimation is a calculated as a function of several factors, including the vertical
mass concentration, temperature-dependent humidity gradients between the snow particles and
the atmosphere, conductive and advective energy- and moisture-transfer mechanisms, particle
size, and solar radiation. The SnowModel-LG blowing snow transport and sublimation fluxes
represent column integrated values in units of kg $m^{-1}$ $s^{-1}$ and cm SWE $d^{-1}$, respectively.
SnowModel-LG variables are output as 3-hourly values on an EASE grid with a resolution of 25
km.
**2.3 Blowing snow sublimation estimates from DY2001**
We also include estimates of the bulk blowing snow sublimation rate ($Q_{bs}$ in Eq. 1) using the
approach described by Déry & Yau (1999, 2001) and subsequently J. Yang & Yau (2007).
Throughout the analysis we refer to this approach as DY2001. We chose to include it because it
is computationally efficient and has been widely applied in studies of blowing snow aerosol
production over sea ice (e.g., Gong et al., 2023; Frey et al., 2020; Huang et al., 2020; Huang &
Jaeglé, 2017; X. Yang et al., 2008, 2019). Sublimation depends on several factors including
surface windspeed, temperature, and humidity deficit.
Following X. Yang et al. (2008), sublimation is scaled by snow age *A'* which accounts for the
reduced ease of wind lofting as snow ages. For a full description of the sublimation calculation
used here, we refer the reader to section 2.1.1 of X. Yang et al. (2008). In our calculations, we
adopt a representative mean snow age of 3 days over Arctic sea ice (Huang & Jaeglé, 2017).
A key factor controlling blowing snow occurrence in DY2001 is the threshold windspeed, which
follows Li & Pomeroy (1997a). The threshold windspeed ($U_t = 6.975 + 0.0033[T_{2m} + 27.27]^2$) is
estimated from the 2 m surface air temperature ($T_{2m}$) and has a minimum value of ~ 7 m $s^{-1}$ at an
air temperature of -27$^o$C. At both higher and lower temperatures, the threshold wind speed will



be larger (maximizing at ~ 10 m s$^{-1}$ for temperatures near 0$^{\circ}$C). We estimate the DY2001
threshold windspeed and blowing snow sublimation using the same meteorology (10 m
windspeed, 2 m temperature, and 2 m RH$_{ice}$) used to derive the ICESat-2 sublimation.

**2.4 ICESat-2 and model gridding procedure**

We aggregate the ICESat-2 observations to a National Snow and Ice Data Center (NSIDC)
Equal-Area Scalable Earth (EASE) grid (Brodzik & Knowles, 2002) with a horizontal resolution
of 100 km. This resolution balances spatial detail with observational coverage, ensuring
sufficient ICESat-2 sampling within each grid cell while minimizing noise that would arise at
finer resolutions due to the narrow swath of the lidar. Temporal resolution is determined by the
duration of the binning period, allowing flexibility to examine daily, seasonal, or multi-year
patterns.
Within each 100 km grid cell, the blowing snow occurrence for a specified time window is
computed as the number of profiles with a blowing snow detection divided by the total number
of valid profiles. A valid profile is defined as one where the surface return is clearly detected,
which excludes profiles with optically thick cloud cover (optical depth > 3), where surface
detection is unreliable or is not achieved. For blowing snow properties such as geometric and
optical depths, only blowing snow retrievals are gridded.
For comparison with model estimates, we extract values from the SnowModel-LG fields (25 km
resolution) by sampling the nearest-neighbor grid point to each valid ICESat-2 profile location.
These sampled values are then binned to the same 100 km EASE grid alongside the ICESat-2
data. We apply the same procedure to the DY2001 estimates: values are first computed at the
location of each valid ICESat-2 profile, and the resulting fields are aggregated onto the 100 km
grid for direct comparison with both ICESat-2 observations and SnowModel-LG outputs.

**2.5 December 2022 example of observed and predicted blowing snow**

Figure 1 highlights a blowing snow storm which occurred over the Central Arctic in December
2022. During an orbit which transited from Svalbard towards the Canadian Arctic Archipelago,
ICESat-2 retrieved blowing snow for roughly 1,200 km along track, with depths up to 250 m and
observed backscatter exceeding $1.50 \times 10^{-4}$ m$^{-1}$ sr$^{-1}$ (Fig. 1a). In this region, MERRA-2
windspeeds ranged from 7.5 to 15 m s$^{-1}$ (blue line, Fig. 1b) and SnowModel-LG predicted
intense blowing snow, with mass fluxes peaking at 4 Mg m$^{-1}$ d$^{-1}$ (green line, Fig. 1b). The
strongest ICESat-2 observed and SnowModel-LG predicted blowing snow occurred coincident
with the strongest winds (middle of Fig. 1a,b). While ICESat-2 did retrieve blowing snow to the
west of this maximum (left side, Fig. 1a) coincident with windspeeds > 8 m s$^{-1}$, SnowModel-LG
predicted only minimal blowing snow mass transport.

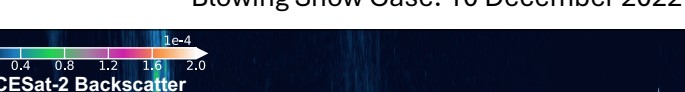

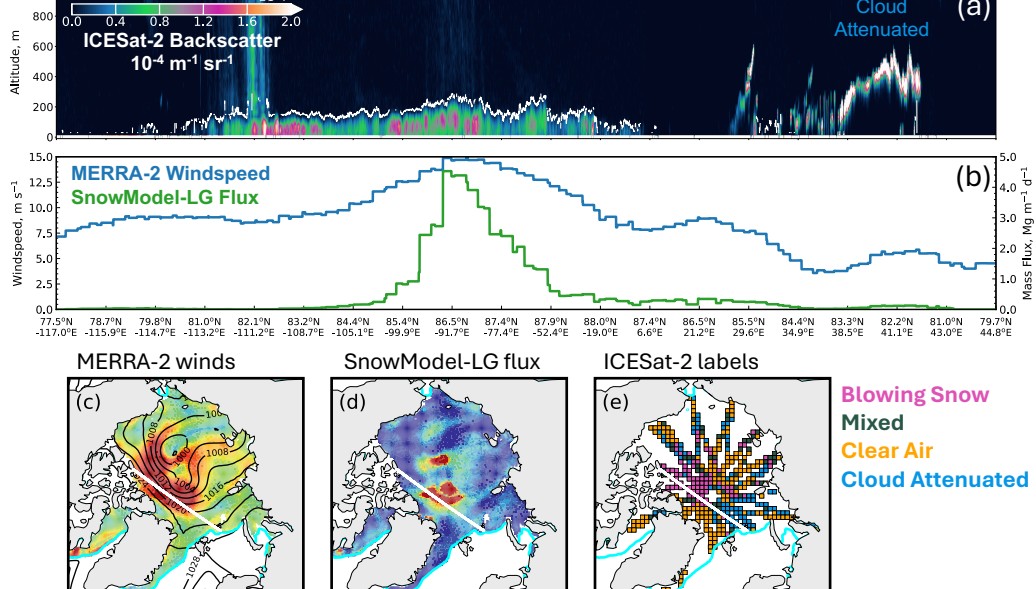

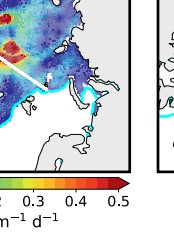

**Figure 1.** Case study of a blowing snow event in the Central Arctic on 10 December 2022. **(a)** ICESat-2 backscatter (shading, units m$^{-1}$ sr$^{-1}$) along an orbit from the Canadian Arctic Archipelago towards north of Svalbard. The white line indicates the top of the blowing snow layer. **(b)** MERRA-2 windspeed (blue line, units m s$^{-1}$) and SnowModel-LG blowing snow mass transport (green line, units Mg m$^{-1}$ d$^{-1}$) along the ICESat-2 orbit shown in panel a. **(c)** Spatial distribution of MERRA-2 windspeed (shading, units m s$^{-1}$) with sea-level pressure contours (black, 4 hPa intervals). **(d)** Spatial distribution of SnowModel-LG blowing snow mass transport (shading, units Mg m$^{-1}$ d$^{-1}$). **(e)** ICESat-2 classifications at 100 km resolution: blowing snow (magenta), mixed (green), clear air (orange), or cloud attenuated (blue) as described in Section 2.5. In panels c-e, the cyan line marks the 15% sea ice contour, while the white line shows the track of ICESat-2 from panel a.

Winds in excess of 8 m s$^{-1}$ covered much of the Central Arctic and coincided with tight sea-level pressure (SLP) gradients stretching from the Beaufort to Lincoln Sea (Fig. 1c). SnowModel-LG predicted blowing snow mass transport > 0.20 Mg m$^{-1}$ d$^{-1}$ over an area of 750,000 km$^2$ (Fig. 1d), which is slightly larger in size than the state of Texas. Given a total Central Arctic area of roughly 3.2 million km$^2$, this storm impacted about a quarter of the basin.

To examine the spatial distribution of ICESat-2 profiles, we first gridded the ICESat-2 orbits to the 100 km grid (Section 2.4) and then assigned each grid cell to one of four categories: blowing snow, mixed, clear air, or cloud attenuated. If more than 70% of all profiles were attenuated due to clouds, the grid cell was labeled as cloud attenuated. We assigned the other three categories based on the occurrence of blowing snow: blowing snow if more than 50% of profiles were blowing snow, mixed if 15-50% of profiles were blowing snow, and clear air if less than 15% of profiles were blowing snow. ICESat-2 grid cells in the western Central Arctic were consistently



classified as blowing snow (magenta colors, Fig. 1e), coinciding with the strongest winds and the
highest SnowModel-LG predicted transport. The total area of ICESat-2 grid cells labeled as
blowing snow was 740,000 km$^2$, closely matching the SnowModel-LG predictions and
confirming that the blowing snow was synoptic in scale, covering much of the Central Arctic.
**3 Blowing snow occurrence frequency and properties from ICESat-2**
**3.1 Spatiotemporal variability and drivers of blowing snow occurrence**
Figure 2 shows the mean multi-year blowing snow occurrence and properties derived from the
ICESat-2 observations for November through April 2018-2023. To generate the average maps,
we grid each cold season independently (following Section 2.4) and then average the five cold
seasons together. We found a significant fraction of the central Arctic experiences blowing snow
frequencies > 25%, with maxima of near 35% in the Fram Strait region (Fig. 2a). This is
consistent with several previous studies which showed these regions have consistent influence (>
15% of the time) from storms entering the Arctic (e.g., Clancy et al., 2022; Valkonen et al.,
2021). This is also evident in the spatial distribution of MERRA-2 windspeeds (Fig. 2d), where
the region of high blowing snow occurrence frequency is collocated with average windspeeds >
6.5 m s$^{-1}$.

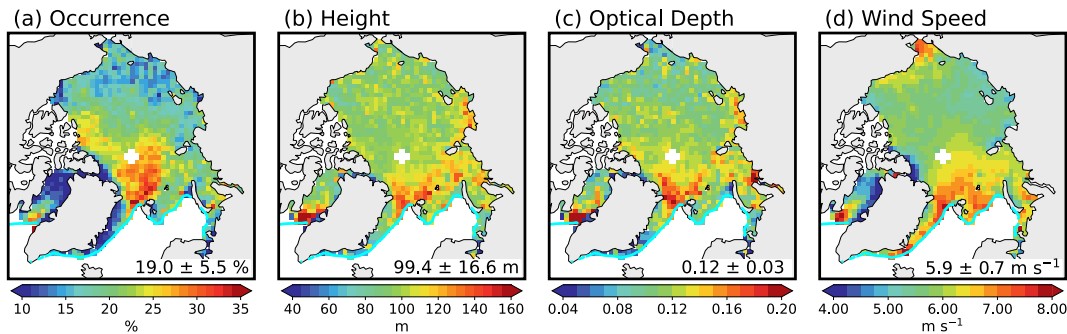

**Figure 2.** Mean ICESat-2 blowing snow properties during the cold season (November-April, 2018-2023): **(a)**
occurrence frequency, **(b)** blowing snow geometrical depth, **(c)** optical depth, and **(d)** MERRA-2 10 m wind speed.
The cyan line marks the 15% sea ice contour. Numbers in the bottom right of each panel correspond to the mean and
standard deviation for values over sea ice.
The ICESat-2 occurrence frequency does not include shallow (< 30 m thick) blowing snow
layers, since these cannot be reliably detected at the vertical resolution of the atmospheric
backscatter profiles. In addition, ICESat-2 cannot sample conditions where optically thick clouds
prevent the surface from being detected. Regions of the Kara, Barents, and Greenland Seas are
particularly susceptible to this under sampling, where the ICESat-2 cloud attenuated occurrence
(% of all profiles where the surface cannot be detected) can exceed 50% across much of the cold
season (Fig. S2).
The multi-year cold season ICESat-2 retrievals show blowing snow layers averaging ~ 100 m in
depth, ranging from ~ 50 m up to 160 m (Fig. 2b). Our previous analysis of ICESat-2
observations near the 2019-2020 MOSAiC campaign demonstrated that low level turbulence
often mixes blowing snow to the top of the surface inversion (Robinson et al., 2025), suggesting





that blowing snow layer depth may serve as a useful indicator of Arctic inversion depth. Blowing
snow optical depths average 0.12 across the Arctic, with maxima near 0.20 in the Fram Strait and
southern Baffin Bay. These regions also experience thicker blowing snow layers on average.
Figure 2 further shows that regions of deeper, optically thicker blowing snow are co-located with
areas of high occurrence frequency and stronger winds.

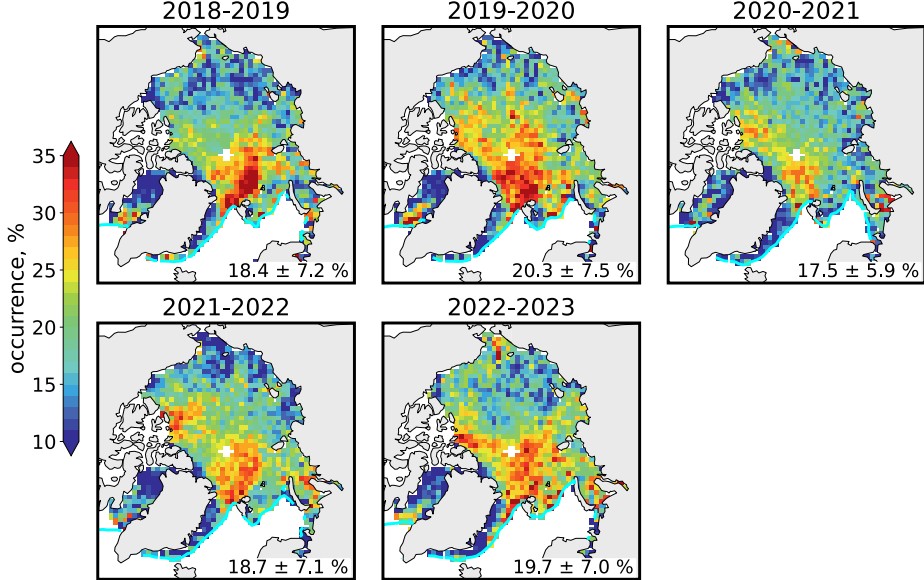

**Figure 3.** Interannual spatial variability of blowing snow occurrence frequency (units %) from ICESat-2
observations during the 2018-2023 cold seasons. Each panel shows the average pan-Arctic occurrence and standard
deviation (lower right). The cyan line marks the 15% sea ice concentration contour.

Figure 3 shows that the ICESat-2 pan-Arctic blowing snow occurrence frequencies are consistent
from year-to-year at 18-20%. The spatial pattern of occurrence also remains fairly consistent,
with the Central Arctic and Fram Strait displaying the highest frequencies and only moderate
shifts in location. Despite this, the Central Arctic can display substantial year-to-year variability.
For example, the highest (2019-2020) and lowest (2020-2021) pan-Arctic frequencies were
observed in consecutive cold seasons.

The contrast between these two cold seasons appears closely aligned with large scale climate and
atmospheric circulation patterns, particularly the Beaufort High and the Arctic Oscillation (AO).
In early 2020, a record positive AO phase (+3.5, top row Fig. S3) coincided with a collapse of
the Beaufort High, enhanced cyclone activity (Ballinger et al., 2021; Rinke et al., 2021), and
widespread blowing snow. From January to March 2020, MERRA-2 sea-level pressure (SLP)
and windspeed featured an elongated region of consistently low pressure (< 1,000 hPa) extending
from Iceland into the ice-covered Kara and Barents Seas (Fig. 4a). Over these regions and the
Central Arctic, mean windspeeds reached 7-9 m s$^{-1}$ (Fig. 4b). During this period, ICESat-2
observed several intense blowing snow episodes covering more than 25% of sea ice area
(blowing snow > 1×10$^6$ km$^2$; Fig. S4), with mean pan-Arctic blowing snow frequencies of
21.9%, reaching up to 50% in the Central Arctic (Fig. 4c).





411

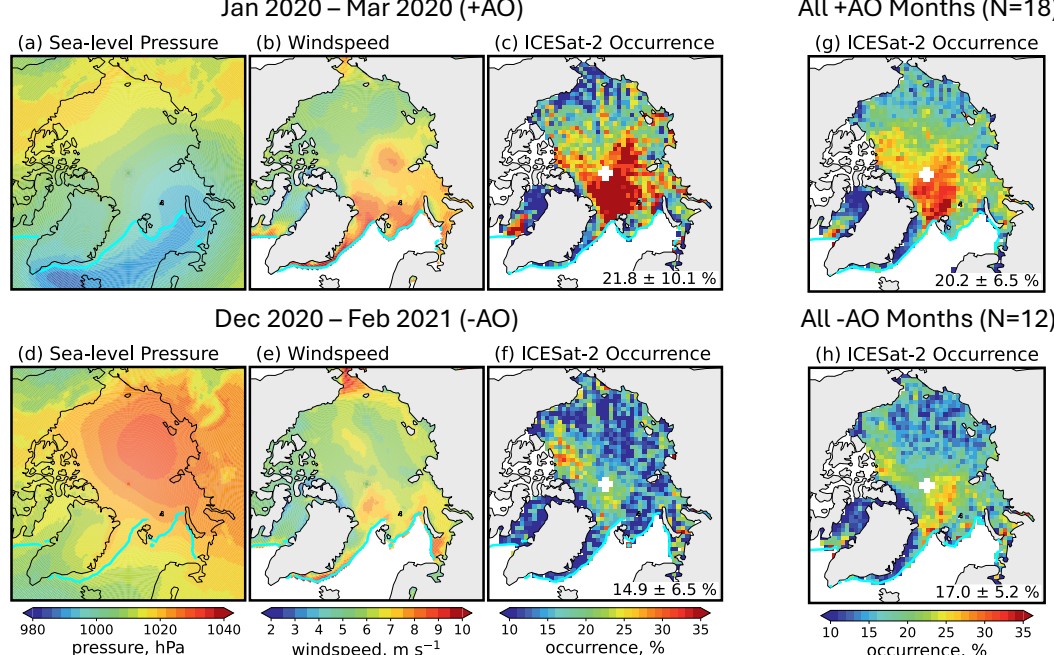

**Figure 4.** Comparison of **(a,d)** MERRA-2 sea-level pressure (hPa), **(b,e)** MERRA-2 wind speed (m s⁻¹), and **(b,d)** ICESat-2 observed blowing snow occurrence frequency (%) for January 2020 – March 2020 (a-c) and December 2020 – February 2021 (d-f). **(g,h)** Composite ICESat-2 blowing snow occurrence frequency for months with (g) positive and (h) negative Arctic Oscillation phases during the 2018-2023 cold seasons.

In contrast, the 2020-2021 season was marked by a strong negative AO (-2.4, top row Fig. S3) and a persistent Beaufort High (mean MERRA-2 SLP > 1,020 hPa across most of the Arctic basin, Fig. 4d), conditions known to suppress storm activity (Kenigson & Timmermans, 2021; Serreze & Barrett, 2011). Consistent with this pattern, MERRA-2 windspeeds were on average ~ 2 m s⁻¹ lower relative to January-March 2020 (Fig. 4e). From December 2020 to February 2021 ICESat-2 detected substantially less blowing snow (47% lower relative to Jan-Mar 2020), with frequencies in the Central Arctic maximizing at only ~ 25% (Fig. 4f). Across all months, we find a moderately strong correlation between AO phase and ICESat-2 blowing snow occurrence (r = 0.62; Fig. S3c). Composites highlight this relationship: positive AO months (N=18; Fig. 4g) exhibit 20% more blowing snow than negative AO months (N=12, Fig. 4h), with particularly large differences (up to a factor of two) in the Fram Strait and Central Arctic.

**3.2 Relationship between windspeed and blowing snow**

In the following section we focus on the Central Arctic region during January-March, the region most well-sampled by ICESat-2 and months least affected by optically thick clouds (Fig. S2). To examine relationships between meteorological factors and blowing snow, we use daily 100 km grid-cell averages. Although this lowers the total number of samples compared to a profile-based approach, averaging helps to reduce noise.




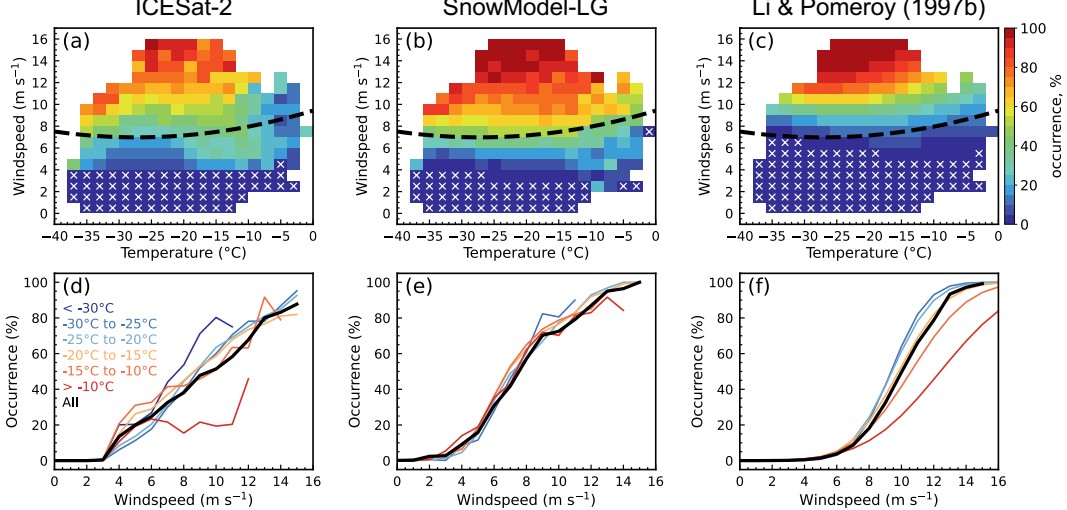

439
440
**Figure 5.** Top row: Dependence of blowing snow occurrence frequency on windspeed and temperature for **(a)**
ICESat-2, **(b)** SnowModel-LG (blowing snow transport fluxes > 0.20 Mg m$^{-1}$ d$^{-1}$), and **(c)** Li & Pomeroy (1997b)
(mean snow age = 72 hours). White stippling represents conditions with no blowing snow; the black dashed line
shows the DY2001 threshold windspeed. Bottom row: Dependence of blowing snow occurrence frequency on
windspeed for all temperatures (black), and for different temperature ranges (< -30ºC; -30 to -25ºC; -20 to -15ºC; -
15º to -10ºC; > -10ºC) for **(d)** ICESat-2, **(e)** SnowModel-LG, and **(f)** Li & Pomeroy (1997b).

Figure 5 compares the blowing snow occurrence as a function of windspeed and temperature.
For comparison to ICESat-2 and SnowModel-LG, the blowing snow occurrence from Li &
Pomeroy (1997b) is also shown (see their Eq. 7). The blowing snow occurrence from Li &
Pomeroy (1997b) is based on a statistical analysis of observations for 16 stations on the prairies
of western Canada and is a function of windspeed, temperature, and snow age (assumed in our
analysis to be 72 hours). It is also in contrast to DY2001, where the threshold windspeed
essentially acts as an on-off switch for blowing snow. ICESat-2 retrievals indicate a 10-40%
blowing snow occurrence below the DY2001 threshold of ~ 7 m s$^{-1}$ (black dashed line, Fig. 5a),
with a much stronger dependence on windspeed than on temperature (Fig. 5a). For example, at 8
m s$^{-1}$, the ICESat-2 occurrence is 50-60 % across all temperatures, while at -25 ºC it rises from
10-15% at 4 m s$^{-1}$ to > 80 % at 15 m s$^{-1}$. SnowModel-LG predictions (defined as blowing snow
transport > 0.20 Mg m$^{-1}$ d$^{-1}$) display frequencies ~10 % larger than ICESat-2 on average but
capture similar features (Fig. 5b). The occurrence of blowing snow predicted from Li & Pomeroy
(1997b) displays a narrower transition region, increasing sharply from < 20 % to > 60 % over the
8-10 m s$^{-1}$ range.

The one-dimensional distributions (Fig. 5d-f) further emphasize the dominant control of
windspeed, with all three datasets showing increasing occurrence with stronger winds. ICESat-2
and SnowModel-LG show a weak temperature dependence, with slightly lower occurrence at
higher temperatures, especially for stronger winds, consistent with enhanced snow cohesion and
bonding resistance (Fig. 5d,e). The Li & Pomeroy (1997b) formulation shows a stronger
temperature sensitivity, ranging from 75% at T < -30 ºC to 20% at T > -5 ºC for a 10 m s$^{-1}$





windspeed (Fig. 5e). The temperature dependence is likely stronger because of our assumption of
a fixed snow age of 72 hours. Snow age also influences bonding and cohesion, with older snow
being more resistant to erosion. Because SnowModel-LG and ICESat-2 sample a range of snow
ages, their apparent temperature dependence is likely weaker.
ICESat-2 blowing snow properties also show a strong dependence on windspeed (Fig. 6a).
Median blowing snow layer height increases from 30 m at windspeeds of ∼ 4 m s$^{-1}$ to more than
150 m at windspeeds > 14 m s$^{-1}$. Optical depth exhibits a similar relationship, rising from 0.02 to
0.26 over the same windspeed range. The spread in both height and optical depth (shading, Fig.
6a) also widens with increasing windspeed, which we attribute to increased noise from fewer
observations in the highest windspeed bins (Fig. 6b).

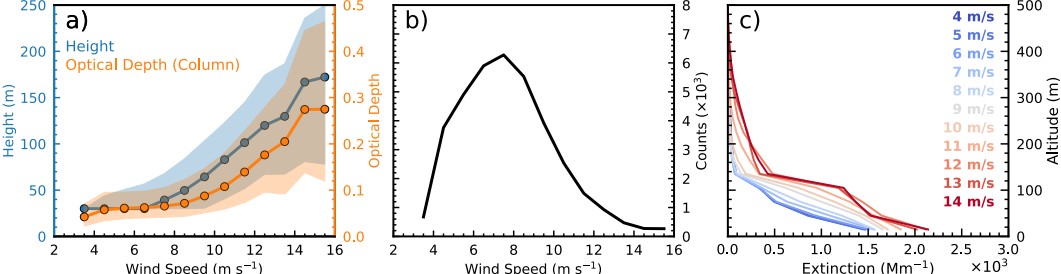

**Figure 6.** Dependence of ICESat-2 blowing snow height and optical depth on windspeed. **(a)** Median (circles with
line) and interquartile range (shading) of ICESat-2 retrieved blowing snow geometric depth (blue) and optical depth
(orange) as a function of 10 m windspeed, using 0.5 m s$^{-1}$ bins. **(b)** Number of grid cells (in thousands) in each
windspeed bin from panel a. **(c)** Mean blowing snow extinction profiles from ICESat-2 retrievals in February 2022
(N = 678,914), grouped in 1 m s$^{-1}$ wide windspeed bins.
The increase in blowing snow optical depth reflects a combination of increased blowing snow
height and stronger backscatter signal (Fig. 5c). Across nearly 700,000 ICESat-2 retrievals in
February 2022, near-surface blowing snow extinction increased by 40% from $1.5 \times 10^3$ Mm$^{-1}$ at 4
m s$^{-1}$ to $2.1 \times 10^3$ Mm$^{-1}$ at 14 m s$^{-1}$. The enhancement is even larger aloft (a factor of 2-3).
Together, these results indicate that stronger winds loft more blowing snow higher into the
atmosphere, consistent with previous studies (Palm et al., 2011, 2018; Robinson et al., 2025).
**4 Contribution of blowing snow to the Arctic snow-on-sea ice budget**
In this section we examine the contribution of blowing snow to the Arctic cold season snow-on-
sea-ice budget. We focus on column integrated blowing snow mass transport ($Q_t$ in Eq. 1) and
sublimation ($Q_{bs}$ in Eq. 1) fluxes, placing them in the context of one another and comparing them
to accumulated snowfall. When interpreting the magnitude of the ICESat-2 estimates, we note
that they depend on assumptions inherent to the backscatter-to-flux conversions (Palm et al.,
2017; Robinson et al., 2025), including prescribed blowing snow particle sizes and the use of
modeled meteorological fields to represent near-surface windspeed, temperature, and humidity
(section 2.1). Blowing snow particle sizes are assumed to decrease exponentially with height,
while sublimation rates increase with higher temperatures and lower humidities.




### 4.1 Blowing snow transport from ICESat-2 and SnowModel-LG

Figure 7 shows the spatial distribution of blowing snow transport flux inferred from ICESat-2. The flux is calculated by combining the ICESat-2 derived mass concentrations with the vertical profile of windspeed, integrated over the depth of the blowing snow layer. The pan-Arctic mean transport flux observed by ICESat-2 is 74 Mg m$^{-1}$, with maxima > 160 Mg m$^{-1}$ in the Central Arctic, co-located with regions of frequent and intense blowing snow (Fig. 1). SnowModel-LG produces a similar spatial distribution but yields transport fluxes that are 2-3 times lower. This discrepancy likely arises because SnowModel-LG confines blowing snow to the lowest several meters of the atmosphere, where winds are weaker. In contrast, ICESat-2 detects blowing snow layers extending several hundred meters above the surface (Fig. 1b, 6a), where stronger winds enhance snow transport. To support this interpretation, we examined the pan-Arctic blowing snow burdens (mass per square meter; Fig. S5) and found that they agree to within about 20% between ICESat-2 (0.17 g m$^{-2}$) and SnowModel-LG (0.14 g m$^{-2}$). In the Central Arctic regions of enhanced transport, both datasets have mean blowing snow burdens of up to 0.40 g m$^{-2}$.

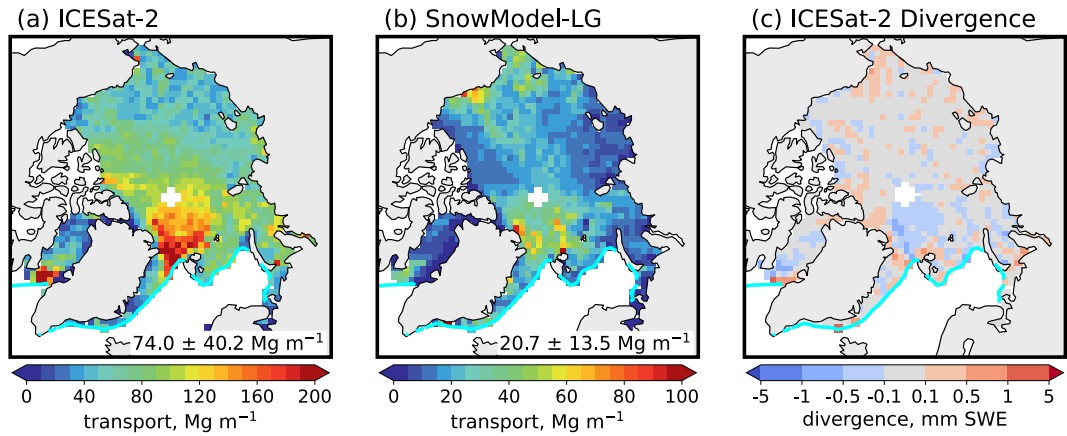

**Figure 7.** Mean 2018-2023 cold season blowing snow transport flux (Mg m$^{-1}$) from **(a)** ICESat-2 and **(b)** SnowModel-LG. Note the color scale for SnowModel-LG is different than for ICESat-2. **(c)** Divergence of blowing snow transport inferred from ICESat-2 (mm SWE).

Although the spatial pattern of transport broadly agrees, our seasonal values are smaller than those reported by J. Yang et al. (2010). Their simulations for December 2006 – February 2007 suggested transport fluxes up to 800 Mg m$^{-1}$ in the Central Arctic and > 1,000 Mg m$^{-1}$ along Greenland's east coast. These higher values could reflect methodological differences: their model did not explicitly account for variable snowpack conditions, which could lead to an overestimate in blowing snow occurrence and transport, and was run at finer spatial (18 km) and temporal (5 s) resolutions, which could capture small-scale wind gradients and localized enhancements in snow redistribution. Despite these differences, both our results and those of J. Yang et al. (2010) indicate that blowing snow transport plays a relatively minor role in the basin-scale snow budget. For example, the divergence of ICESat-2 transport (Fig. 7e) is limited to a few tenths of mm SWE, with localized maxima near 1 mm SWE in regions of frequent blowing snow.





**4.2 Multi-year estimates of blowing snow sublimation**
Figure 8 shows the mean total annual blowing snow sublimation and snowfall for the 2018-2023
cold seasons. Pan-Arctic blowing snow sublimation totals from ICESat-2 (1.63 cm SWE) are in
close agreement with SnowModel-LG (1.66 cm SWE) and within 30% of DY2001 (2.07 cm
SWE). All three estimates are broadly consistent with previous modeling studies (Chung et al.,
2011; Liston et al., 2020; J. Yang et al., 2010). In the Central Arctic near Svalbard, ICESat-2
indicates the highest values of sublimation (3-4 cm SWE). A secondary maximum (> 3 cm SWE)
occurs in the Barents Sea, where blowing snow is retrieved half as often. This reflects the
sensitivity of sublimation to temperature and humidity, because the marginal seas are generally
warmer than the Central Arctic (Fig. S6). Thus, the reduced occurrence of blowing snow is offset
by higher temperatures and lower humidity, which enhance sublimation.
We compare blowing snow sublimation to total MERRA-2 snowfall over the cold season (12.41
cm SWE, Fig. 8d). On average, we find that blowing snow removes 13.6% (ICESat-2), 14.1%
(SnowModel-LG), and 16.9% (DY2001) of snowfall. The regional impact, however, varies
strongly (Figs 8e-g). In the Kara and Barents Seas, where snowfall is highest, sublimation
removes only 5-10% of snowfall. In the Central Arctic losses increase to 18-24%, while in
regions with more moderate snowfall, such as the Beaufort Sea, sublimation losses can exceed
30% (e.g., 2-3 cm SWE of sublimation compared to 8-10 cm SWE of snowfall).
The fraction of snowfall removed by blowing snow sublimation inferred from ICESat-2 reaches
30% in the Beaufort Sea north of the Canadian Arctic Archipelago (Fig. 8e). SnowModel-LG
and DY2001 show a similar enhanced offset, though their maxima are shifted southeastward
along the coast of Alaska (Fig. 8f,g). The 2018-2023 period was marked by several strong
Beaufort High episodes, such as the 2021-22 event highlighted in Fig. 4 (NSIDC, 2021), which
are typically associated with calm, dry conditions. Under such conditions, ICESat-2 retrievals
may occasionally overestimate blowing snow. False positives could arise when low-level ice
crystals (ice clouds or diamond dust) mix with blowing snow, leading the entire ICESat-2
backscatter signal to misattributed to blowing snow. This effect was most pronounced during
winter 2021-2022, when exceptionally warm (T > -20ºC) and dry (RH$_{ice}$ < 90%) conditions
prevailed north of the Canadian Arctic Archipelago (Fig. S7).



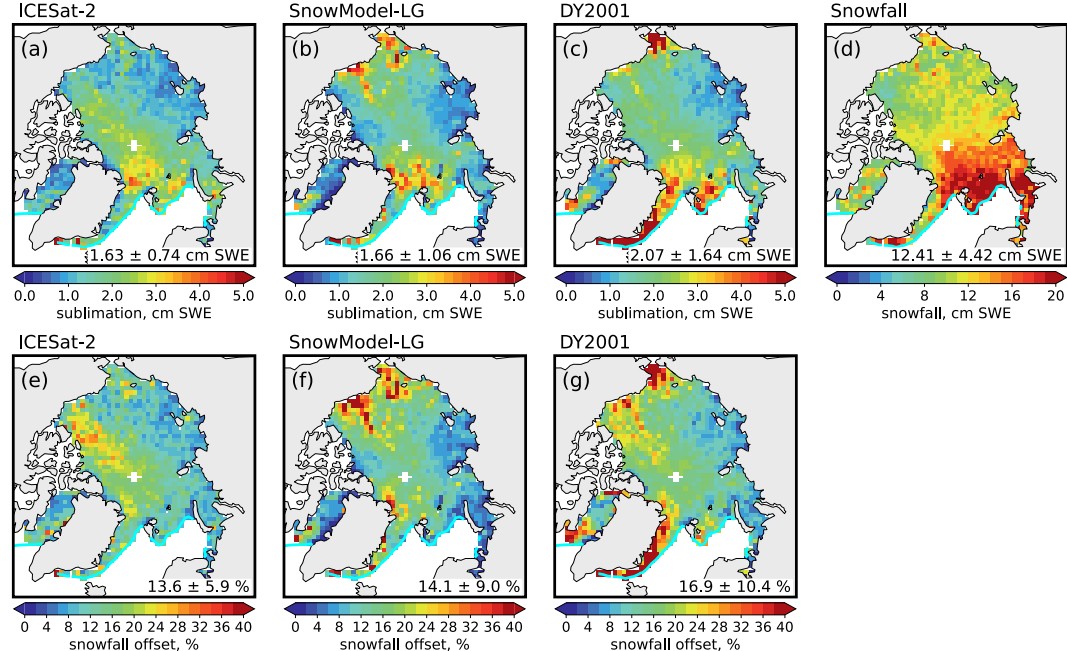

**Figure 8.** Spatial distribution of blowing snow sublimation, total snowfall, and the contribution of blowing snow sublimation to snowfall offset over Arctic sea ice during 2018-2023. **(a-c)** Total blowing snow sublimation (cm SWE) inferred from (a) ICESat-2, (b) SnowModel-LG, and (c) DY2001. **(d)** Total MERRA-2 snowfall (cm SWE). **(e-g)** Percent of snowfall removed by blowing snow sublimation (= 100 × [sublimation / snowfall]) from (e) ICESat-2, (f) SnowModel-LG, and (g) DY2001.

Along Greenland's east coast, DY2001 predicts much higher sublimation fluxes (4-5 cm SWE, > 70% of snowfall) than either ICESat-2 and SnowModel-LG (2-3 cm SWE, 20-30% of snowfall). This discrepancy likely reflects DY2001's simple threshold-based parameterization, which tends to overpredict blowing snow at the typical windspeeds in this region (6-8 m s$^{-1}$, Fig. 1). Warmer and drier conditions in this region (Fig. S6) further amplify the sublimation predicted by DY2001.

Daily pan-Arctic time series (Fig. 9) show that blowing snow sublimation is nearly continuous throughout the cold season, punctuated by sharp peaks during major storm events. The most intense episodes (> 0.04 cm SWE d$^{-1}$ averaged over sea ice) occur only a few times per season and correspond to widespread blowing snow detected by ICESat-2 (Fig. S4). These storms contribute disproportionately to the seasonal total, with individual events removing up to 60% of daily snowfall (Fig. S7). Between storms, sublimation persists at lower but steady rates (0.01-0.02 cm SWE d$^{-1}$) and these background losses accumulate to a substantial share (35-40%) of the seasonal total.





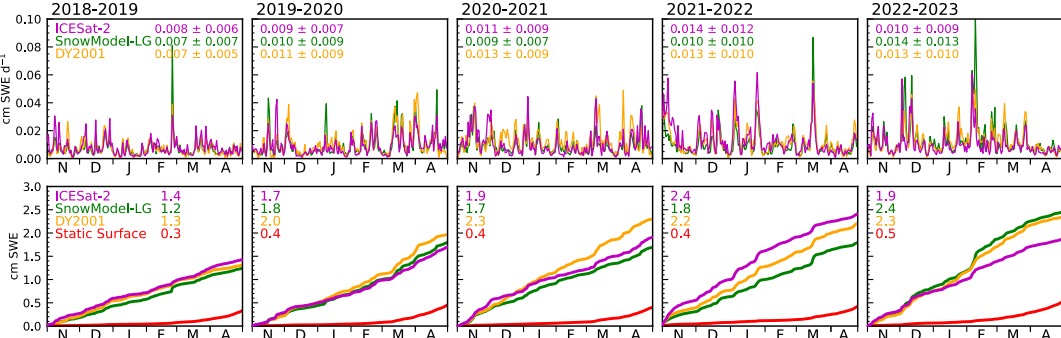

**Figure 9.** Timeseries of blowing snow sublimation across five Arctic cold seasons. **(top row)** Daily blowing snow sublimation (cm SWE d$^{-1}$) inferred from ICESat-2 (magenta line) and predicted by SnowModel-LG (green line) and DY2001 (orange line). **(bottom row)** Cumulative daily blowing snow sublimation (cm SWE). The red lines in the bottom row represent the cumulative static surface (non-blowing snow) sublimation predicted by SnowModel-LG.

The ICESat-2 inferred sublimation ranges from 1.4 to 2.4 cm SWE across the five cold seasons (Fig. 9, bottom row), corresponding to a 11-20% offset of seasonal snowfall. Both snowfall and blowing snow sublimation vary by 1-2 cm SWE year to year, but the two do not always covary. For example, the 2021-2022 cold season had the lowest snowfall (11.9 cm SWE) yet the highest ICESat-2 sublimation (2.4 cm SWE, 20% offset). Conversely, 2018-2019 featured higher snowfall (12.9 cm SWE) but relatively low sublimation (1.4 cm SWE, 11% offset). These interannual differences highlight that sublimation depends not only on storm frequency and strength (which also drive snowfall) but also on atmospheric conditions which regulate blowing snow occurrence and sublimation efficiency. SnowModel-LG and DY2001 generally agree with the ICESat-2 sublimation, though DY2001 tends to predict slightly higher values.

Blowing snow sublimation exceeds surface sublimation by a factor of 4-5, underscoring the dominant role of blowing snow in sublimation-driven snow loss during much of the cold season. The cumulative surface sublimation timeseries ($Q_{ss}$ in Eq. 1) predicted by SnowModel-LG is shown in Fig. 9 (red lines, bottom row). Seasonal total surface sublimation averages only 0.3-0.5 cm SWE, with nearly all of it occurring from late February through April, when solar radiation increases, near-surface air warms, and RH$_{ice}$ decreases. These values are lower than the 1-2 cm SWE reported by Déry and Yau (2002), likely because their annual means included the warmer spring and summer months. Consistent with this, SnowModel-LG calculates an Arctic-wide annual mean surface sublimation of ~ 1 cm SWE.

ICESat-2 likely underestimates blowing snow sublimation because it cannot observe blowing snow beneath optically thick clouds. These conditions are most frequent during winter storms, when strong winds can drive intense sublimation. To assess this sampling bias, we examine the 2018-2023 SnowModel-LG and DY2001 predictions under all conditions (i.e., regardless of whether ICESat-2 detected the surface). The all-conditions maps (Fig. S9) show patterns similar to Fig. 8 but with magnitudes 16-25% larger. Pan-Arctic blowing snow sublimation totals increase to 2.1 cm SWE for SnowModel-LG and 2.4 cm SWE for DY2001. Comparing these values to the seasonal snowfall from Fig. 8 (12.4 cm SWE) yields offsets of 17% for SnowModel-LG and 19% for DY2001. This comparison suggests that ICESat-2 captures the



spatial pattern and temporal variability of blowing snow sublimation well but underestimates the
total by roughly 20% due to this sampling bias.

**5 Summary and conclusions**

We presented the first multi-year pan-Arctic estimates of blowing snow derived from ICESat-2
satellite observations, extending our earlier single-year analysis (Robinson et al., 2025) to five
cold seasons (November through April, 2018-2023). ICESat-2 retrievals allowed us to
characterize blowing snow occurrence and properties (geometric and optical depths), and, when
combined with assumptions about particle sizes and meteorology from reanalysis, to infer
blowing snow sublimation and evaluate its contribution to the Arctic snow-on-sea ice budget.

Over the five seasons analyzed, ICESat-2 retrievals indicate a mean pan-Arctic blowing snow
occurrence of 19%, with maxima exceeding 30% in the Central Arctic and Atlantic sector,
regions frequently impacted by storms arriving from lower latitudes. Retrieved blowing snow
geometric and optical depths also maximize in these regions. Interannual variability of blowing
snow occurrence is substantial and is driven by the Arctic Oscillation (AO). We find that positive
AO periods have lower SLP and higher winds, and ~ 50% more blowing snow than negative AO
periods. In the Central Arctic, blowing snow occurrence during the positive AO phase was more
than twice that of the negative phase, a pattern consistent across all five seasons.

ICESat-2 observations confirm that windspeed is the primary driver of blowing snow
occurrence, with temperature acting as a secondary modulating factor. Blowing snow occurrence
increases with windspeed across all temperatures, exceeding 80% at 12 m s$^{-1}$. The physics-based
threshold windspeed in SnowModel-LG (4-5 m s$^{-1}$) is 2-3 m s$^{-1}$ lower than in DY2001 ($\sim$ 7 m s$^{-1}$)
and aligns more closely with the windspeeds at which ICESat-2 reliably detects blowing snow.
Both ICESat-2 and SnowModel-LG suggest blowing snow occurrence frequencies of 10-40% at
windspeeds 4-7 m s$^{-1}$, where DY2001 predicts no blowing snow. Windspeed also strongly
controls blowing snow height and optical depth: blowing snow heights increase from 30 m at 4
m s$^{-1}$ to almost 200 m at 15 m s$^{-1}$, while optical depths rise from 0.02 to 0.26 over the same
range, driven by enhanced backscatter over deeper heights.

Maximum mass transport fluxes peak were blowing snow is most frequent, with seasonal means
of 74 Mg m$^{-1}$ for ICESat-2 and 21 Mg m$^{-1}$ for SnowModel-LG. This factor of three difference
reflects SnowModel-LG's confinement of blowing snow to the lowest few meters, where winds
are weaker, while ICESat-2 detects layers extending to several hundreds of meters, where
stronger winds drive greater transport. Yet, pan-Arctic burdens agree within ~20% (0.17 g m$^{-2}$
for ICESat-2 vs. 0.14 g m$^{-2}$ for SnowModel-LG), underscoring that while the vertical extent is
different, the overall mass is consistent. Despite high transport, divergence in ICESat-2 inferred
mass flux contributes minimally to the snow budget (maximum of 1 mm SWE).

We find that blowing  snow sublimation plays an important role in the Arctic snow-on-sea-ice
budget, reaching up to 5 cm SWE in the Central Arctic, and averaging 1.63-2.07 cm SWE over
all sea ice. This is equivalent to a 13.6-16.9% removal of seasonal snowfall on average, with as
much as 30% removal in some regions such as the Beaufort Sea. The pan-Arctic ICESat-2
inferred blowing snow sublimation ranged from 1.4 to 2.4 cm SWE (11-20% snowfall offset)



across the five cold seasons, with similar estimates from SnowModel-LG (1.2-2.4 cm SWE) and
DY2001 (1.3-2.3 cm SWE). SnowModel-LG and DY2001 predictions under all conditions (i.e.,
including those without ICESat-2 observations due to sampling or clouds) suggest pan-Arctic
blowing snow sublimation could be ~20% larger (2.1-2.4 cm SWE) than was found using
ICESat-2, resulting in a larger snowfall removal of 17-19%. SnowModel-LG indicates that
sublimation from blowing snow is up to a factor of five larger than surface sublimation, which
offsets only an additional 2-4% of snowfall.
Our analysis is limited by a number of factors, including the sampling pattern of ICESat-2.
While the high resolution of atmospheric backscatter allows unprecedented detail into blowing
snow, the narrow spatial sampling requires temporal and spatial averaging, such as binning the
ICESat-2 profiles to a 100 km grid, to generate meaningful statistics. This approach improves
coverage but smooths fine-scale variability and may underrepresent short-lived or localized
blowing snow events. Moreover, the blowing snow algorithm cannot detect blowing snow layers
thinner than 20-30 m. Such thin drifting and blowing snow layers are often predicted by
SnowModel-LG and DY2001. Nevertheless, these discrepancies in vertical resolution and
sampling appear to have a minimal net effect on the overall estimates of blowing snow fluxes,
which are similar for all three methods. Our transport and sublimation flux estimates rely on
reanalysis meteorology, which has been shown to have biases, particularly at high latitudes (e.g.,
Jonassen et al., 2019; Marshall et al., 2018), and currently does not include feedbacks from
blowing snow on the temperature and moisture fields. Such feedbacks would tend to suppress
sublimation by increasing humidity and cooling the near-surface atmosphere, potentially leading
to overestimation of sublimation in our analysis. However, work done on Antarctic blowing
snow processes indicates that the entrainment of warmer and drier air present above the blowing
snow and surface temperature inversion can reduce or even eliminate this sublimation-humidity
feedback (Palm et al., 2018). Incorporating these processes into coupled models would improve
the realism of both meteorological forcing and snow-atmosphere interactions.
Beyond its role in the snow-on-sea-ice budget, blowing snow sublimation also acts as a
significant source of moisture and a sink of heat for the atmosphere. The fate of this moisture
remains poorly constrained and warrants further study. Blowing snow sublimation over sea ice is
also a recognized source of sea salt aerosols (e.g., Frey et al., 2020; Gong et al., 2023; Huang &
Jaeglé, 2017; Ranjithkumar et al., 2025). Taken together, these points highlight that blowing
snow has the potential to impact a range of polar processes including boundary layer structure,
cloud formation and lifetime, atmospheric chemistry, and the surface energy balance. Recent
modeling efforts are beginning to account for these processes (e.g., Hofer et al., 2021; Luo et al.,
2021), offering new opportunities to improve predictions of Arctic composition, weather, and
climate. Such advancements will require robust observational constraints to ensure realism and
guide a process-based understanding of the coupled Arctic system. By capturing the vertical and
horizontal structure of blowing snow at unprecedented scales, our study demonstrates that
spaceborne lidar is a key tool for bridging the gap between observations and models, and for
advancing our understanding of the rapidly changing Arctic environment.



**Code and data availability**

The ICESat-2 ATL09 data used in this study can be accessed through the NASA NSIDC Distributed Active Archive Center (https://doi.org/10.5067/ATLAS/ATL09.006). The code and data required to reproduce the figures in this study are available at: https://doi.org/10.5281/zenodo.18119606.

**Author contributions**

JR and LJ designed the study. SPP aided in ICESat-2 software development and visualization. GEL developed the SnowModel-LG code. JR and LJ performed formal analysis. JR prepared the manuscript with contributions from all co-authors.

**Competing interests**

The authors declare that they have no conflict of interest.

**Acknowledgments**

The authors express gratitude to the ICESat-2 engineering and science teams for their ongoing efforts to maintain the ATLAS instrument and generate the ICESat-2 atmospheric data products.

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
