# Peer review of "Blowing snow contributions to the Arctic snow-on-sea ice"

_EGUsphere, 2026_

## Referee Comment (RC1)

Review of the manuscript by Robinson et al. titled *"Blowing snow contributions to the Arctic snow-on-sea ice budget using ICESat-2 observations"*

**General comments:**

This study is a follow-up to Robinson et al. (2025), who introduced a new method for deriving Arctic blowing-snow properties from ICESat-2 satellite observations and validated it against MOSAiC observational data. In this work, the authors introduce several refinements to their original algorithm to improve the signal-to-noise ratio for pan-Arctic blowing-snow detection. Furthermore, they present the first multi-year (2018–2023) observational estimates of blowing-snow occurrence, mass transport, and sublimation fluxes for the entire Arctic winter season. To support their findings, these estimates are compared with two different model simulations, SnowModel-LG and PIEKTUK. The manuscript presents a novel approach for detecting large-scale blowing-snow properties in the Arctic during the winter season, where in situ field data are scarce, hindering model evaluation and climate impact assessment. In general, the work is well within the scope of TC and well written. I recommend its publication in this journal after considering my points below.

**Specific comments:**

My major concern is that the study treats the pan-Arctic as a single, uniform ice body, without distinguishing between multi-year and first-year ice. It would be more informative if the authors separated the blowing-snow effects across these ice types. I raise this point because a recent study by Li et al. (2025; https://egusphere.copernicus.org/preprints/2025/egusphere-2025-4601/#discussion) suggests that multi-year sea ice may be an important source of sea salt (via blowing snow as proposed) for Svalbard halogen chemistry. From the derived blowing-snow occurrences shown in your Figures 2 and 3, areas of higher blowing-snow frequency appear to coincide largely with regions dominated by multi-year ice (see Fig. 1 of Li et al. or other sea-ice type maps). This overlap is an interesting feature that merits further discussion. With access to ice-type data, the authors could also assess the impact of blowing-snow sublimation on the surface snow mass budget separately for different ice types, similar to the analysis already conducted for the entire Arctic.

Regarding blowing-snow sublimation modelling (lines 103-105), a recent study by Huang et al. (2025) suggests that snow particle fragmentation within the saltation layer may significantly influence the sublimation flux.
 Huang, N., Bao, J., Yu, H., and Li, G.: Snow particle fragmentation enhances snow sublimation, Atmos. Chem. Phys., 25, 12535–12548, https://doi.org/10.5194/acp-25-12535-2025, 2025.

Re Fig. 1. It would be helpful to include the blowing-snow sublimation flux along the track (together with air temperature and RH_ice curves), along with some discussion.

**Technical items:**

Line 470: Fig. 5e must be Fig. 5f.

Line 491: Fig. 5c must be Fig. 5c.

Line 519: Fig. 1b should be 1a?

Line 540: Fig. 7e should be fig. 7c?